# No Retraining at Edge: Efficient Resource-Aware Mixed-Precision Quantization via Federated Supernet Learning

**Lianbo Ma** [1]  **Yonghui Su** [1]  **Nan Li** [2]  **Xingwei Wang** [3]

## Abstract

Federated learning (FL) enables collaborative training across distributed edge devices, but deploying lightweight models in dynamic edge environments remains challenging. Existing methods typically require retraining whenever device resource constraints change, resulting in excessive computational overhead. We propose DFMPQ, a dynamic federated mixed-precision quantization framework that enables retraining-free deployment at the edge. DFMPQ trains a weight-sharing mixed-precision supernet via FL, which jointly represents diverse bit-width configurations. After training, resource-aware quantized subnets can be derived on demand to satisfy heterogeneous and time-varying resource constraints without additional optimization. However, optimizing such a supernet in federated settings is difficult due to optimization interference among heterogeneous bit-widths and the coupling of quantization noise with non-IID data. DFMPQ addresses these issues through semantic-aware training and aggregation mechanisms that stabilize supernet optimization. In addition, a sensitivity-guided greedy search strategy is adopted to efficiently identify suitable quantization configurations under given resource budgets. Extensive experiments on multiple datasets and network architectures demonstrate that DFMPQ achieves competitive accuracy with significantly reduced computational cost, enabling efficient deployment for dynamic edge computing environments.

---

[1]College of Software, Northeastern University, Shenyang, China [2]Shanxi University Institute of Big Data Science and Industry, Taiyuan, China [3]School of Computer Science and Engineering, Northeastern University, Shenyang, China. Correspondence to: Nan Li <linan10@sxu.edu.cn>, Xingwei Wang <wangxw@mail.neu.edu.cn>.

*Proceedings of the 43rd International Conference on Machine Learning*, Seoul, South Korea. PMLR 306, 2026. Copyright 2026 by the author(s).

## 1. Introduction

Federated learning (FL) has emerged as a fundamental paradigm for edge computing, enabling collaborative model training across distributed devices while preserving data privacy (Khouas et al., 2024). By leveraging client-side data, FL facilitates the construction of more generalized global models. However, the limited computational resources of edge devices pose significant challenges, as deploying over-parameterized full-precision models incurs substantial computation and storage overhead (Xu et al., 2024). Such models often lead to prohibitively slow inference, particularly for latency-sensitive applications (Liu et al., 2023). To alleviate these limitations, Mixed-Precision Quantization (MPQ) has been widely adopted as an effective technique for efficient model deployment on resource-constrained devices (Dong et al., 2019; Tang et al., 2023).

MPQ aims to balance model accuracy and system constraints such as model size and inference latency by assigning heterogeneous bit-widths to different network layers. Higher precision is preserved for sensitive layers, while lower precision is applied to less critical ones. Most existing MPQ methods follow a two-stage paradigm (Tang et al., 2024; Guo et al., 2020). In the first stage, bit-width allocation is formulated as a constrained optimization problem to determine an appropriate precision configuration under given resource budgets. In the second stage, the model is quantized accordingly and retrained to compensate for accuracy degradation induced by quantization.

Despite its effectiveness in centralized settings, extending MPQ to federated learning remains challenging. ① The retraining process required to mitigate quantization noise often incurs computational costs comparable to or even exceeding those of federated training itself, imposing an unsustainable burden on resource-constrained edge devices. ② Device heterogeneity further complicates model deployment, as client devices exhibit substantial differences in hardware capabilities. ③ Moreover, device resources such as battery level and thermal condition vary dynamically over time, requiring on-the-fly adaptation of model execution. Designing quantized models in advance for all possible resource states would therefore necessitate maintaining an intractably large pool of pre-trained models.

To address the above challenges, we propose a *Dynamic Federated Mixed-Precision Quantization* (DFMPQ) framework that leverages federated learning to train a weight-sharing supernet encompassing diverse quantization configurations. This design enables the rapid derivation of deployable quantized networks under dynamic resource constraints without additional retraining. Specifically, DFMPQ samples resource-adaptive quantized subnets for local training and subsequently aggregates them into a global supernet. However, optimizing such a system introduces two critical challenges that hinder convergence. First, the optimization trajectories associated with different bit-width configurations often diverge significantly, while the superposition of heterogeneous quantization noise further amplifies interference during weight-sharing optimization. Second, in federated settings with non-independent and identically distributed data, feature drift naturally arises between local models and the global model. When combined with heterogeneous quantization configurations, this feature drift is substantially exacerbated by varying levels of quantization noise.

To mitigate these issues, we propose *Class-Conditional Semantic Alignment* (CCSA). By employing linear Maximum Mean Discrepancy (MMD), CCSA introduces a unified optimization objective in the feature space, explicitly constraining local quantized representations to align with global full-precision semantic prototypes. This mechanism not only corrects semantic drift caused by the coupling of non-IID data and quantization noise, but also provides consistent optimization guidance across heterogeneous bit-width configurations. Furthermore, we design a *Semantic-Aware Hybrid Aggregation* (SAHA) strategy. By evaluating semantic consistency between local client models and the global supernet at critical network stages, SAHA identifies updates that remain corrupted by severe feature drift or quantization noise. The aggregation weights of such updates are then dynamically adjusted, thereby protecting the global supernet from destabilizing contributions. Finally, to efficiently explore the quantization configuration space during deployment, we propose a *Sensitivity-Aware Greedy Search* (SAGS) algorithm. By leveraging gradient-based sensitivity priors, SAGS enables rapid identification of suitable quantization configurations under given resource budgets, significantly accelerating deployment. The main contributions of this work can be summarized as follows:

- We identify and analyze two fundamental challenges that hinder the convergence of weight-sharing mixed-precision supernets in FL, namely quantization-exacerbated feature drift and optimization interference among heterogeneous bit-width configurations.

- We propose CCSA, which introduces a linear MMD-based regularization term to guide client-side training and align local quantized features with global full-

precision semantic prototypes.

- We design a fine-grained SAHA strategy that dynamically regulates aggregation weights based on semantic consistency at critical network stages, thereby stabilizing global supernet optimization.

- Extensive experiments on multiple representative models and datasets demonstrate the effectiveness and robustness of the proposed DFMPQ framework.

## 2. Preliminaries

**Quantization.** One of the most effective ways to quantize neural networks is Quantization-Aware Training (QAT). During the forward pass, the quantization function $Q_b(\cdot)$ takes the input vector $x$ and outputs the quantized vector $\hat{x}$, which is expressed as follows:

$$\hat{x} = Q_b(x; \gamma_b) = \gamma_b \cdot \text{clip}(\lfloor \frac{x}{\gamma_b} \rceil, n, p), \quad (1)$$

where $\lfloor \cdot \rceil$ is the rounding-to-nearest function, $\gamma_b$ is the trainable scaling factor, and $\text{clip}(\cdot, \alpha, \beta)$ is the clipping function, which ensures that the input values fall within the range $[\alpha, \beta]$, and $n$ and $p$ the lower and upper quantization thresholds. Since the quantization function $Q_b(\cdot)$ is non-differentiable, Straight-Through Estimation (STE) is employed to ensure the normal backpropagation of gradients and the update of parameters.

**Federated Learning.** Consider FL system consisting of an edge server and $K$ distributed clients. Each client $i$ holds a private dataset $\mathcal{D}_i = \{(x_j, y_j)\}_{j=1}^{d_i}$, where $(x_j, y_j)$ denotes the feature-label pair of the $j$-th sample, and $d_i$ is the number of local samples on client $i$. The aggregate dataset of the FL system is $\mathcal{D} = \{\mathcal{D}_1, \ldots, \mathcal{D}_K\}$, with total number of samples $N = \sum_{i=1}^{K} d_i$. The local data distributions $\mathcal{D}_i$ may be non-IID, introducing heterogeneity across clients.

In this setting, clients collaboratively train a global model by performing local updates on their private data and sending the updated parameters or gradients to the server for aggregation. The optimization objective of the FL system is to minimize the global loss function $\mathcal{L}(w)$ over all client data, and simultaneously obtain the optimal global model parameters $w^*$:

$$w^* = \arg\min_w \mathcal{L}(w) = \arg\min_w \sum_{i=1}^{K} \frac{d_i}{N} \mathcal{L}_i(w_i; \mathcal{D}_i), \quad (2)$$

where the local loss function on client $i$ is defined as

$$\mathcal{L}_i(w_i; \mathcal{D}_i) = \frac{1}{d_i} \sum_{j=1}^{d_i} \ell(x_j, y_j \mid w_i), \quad (3)$$

with $\ell(\cdot)$ denoting the sample-wise loss function. Here, $w_i$ represents the parameters of the local model on client $i$.

## 3. Problem Statement

**Problem 1. Optimization interference between heterogeneous bit-width configurations in weight-sharing quantized networks.** We illustrate this phenomenon through a 2D regression experiment. As shown in Figure 1a, when simultaneously optimizing a weight-sharing network with 2-bit and 4-bit configurations, the latent weights fail to converge to the global optimum due to inter-bit-width interference, instead exhibiting persistent oscillation. This issue severely impedes convergence speed and, in extreme cases, leads to optimization collapse (Nagel et al., 2022; Ma et al., 2024). Figure 1b depicts the cumulative gradient effect, exposing a stark divergence between the optimization trajectories of 2-bit and 4-bit configurations. Consequently, the effective update direction resulting from the superposition of these conflicting gradients represents a suboptimal compromise that limits the performance of all subnets.

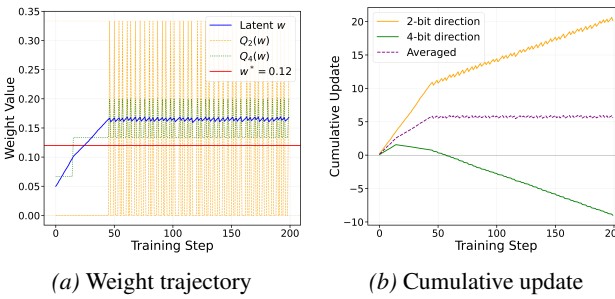

*(a)* Weight trajectory      *(b)* Cumulative update

*Figure 1.* 2D regression experiments on weight-sharing quantized networks with 2-bit and 4-bit configurations. (a) illustrates the oscillation of latent weights caused by optimization interference between heterogeneous bit-width configurations. (b) demonstrates the existence of gradient direction conflicts between these heterogeneous configurations.

**Problem 2. The superposition of quantization noise exacerbates feature drift.** In standard Federated Learning scenarios characterized by heterogeneous (non-IID) data, local models tend to overfit their skewed data distributions. This causes their learned feature representations to deviate from the global semantic consensus, a phenomenon known as Feature Drift (Karimireddy et al., 2020). In our framework, where clients sample and train mixed-precision models satisfying their specific resource constraints, we observe that this phenomenon is significantly exacerbated, as illustrated in Figure 2. The reasons for this amplification are twofold. First, quantization acts as a lossy compression mechanism that results in the loss of fine-grained information within feature maps. Consequently, when a model, whose expressive capacity is diminished by quantization, attempts to fit its biased local data, it may be compelled to

learn biased or erroneous feature representations. Second, the noise introduced by quantization couples with the statistical bias arising from non-IID data. The optimization process is thus burdened with simultaneously minimizing task loss on biased data and compensating for quantization errors. This often drives model weights into sharp, undesirable local minima, causing the final feature extractor to generate highly distorted and erroneous representations.

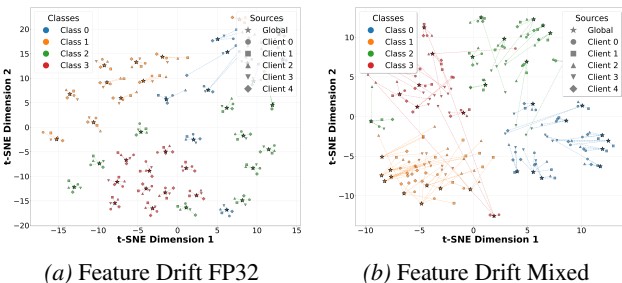

*(a)* Feature Drift FP32      *(b)* Feature Drift Mixed

*Figure 2.* t-SNE visualization of feature drift between client and server models (penultimate layer of ResNet18 on CIFAR10). (a) Clients train full-precision networks. (b) Clients train randomly sampled mixed-precision networks.

## 4. Methodology

As illustrated in Figure 3, the proposed DFMPQ framework operates in two phases: supernet training and deployment. During training stage, we incorporate CCSA and SAHA strategies to ensure robust convergence. Subsequently, in deployment stage, we leverage the SAGS strategy to efficiently locate optimal quantized networks that satisfy client-specific resource constraints.

### 4.1. Class-Conditional Semantic Alignment

Formally, the squared Maximum Mean Discrepancy (MMD) between distributions $P$ and $Q$ in a Reproducing Kernel Hilbert Space (RKHS) $\mathcal{H}$ is defined as follows:

$$\text{MMD}^2(P, Q) = \left\| \mathbb{E}_{z_p \sim P} \left[ \phi(z_p) \right] - \mathbb{E}_{z_q \sim Q} \left[ \phi(z_q) \right] \right\|_{\mathcal{H}}^2 \tag{4}$$

Computing pairwise MMD among $N$ clients incurs a prohibitive $O(N^2)$ complexity. To alleviate this, we propose aligning local features exclusively with Global Prototypes derived from the full-precision supernet, which reduces the complexity to $O(N)$ and preserves privacy by eliminating raw data exchange. Furthermore, while the standard RBF Kernel ($k(x, y) = \exp(-\|x - y\|^2 / 2\sigma^2)$) enforces strict moment matching, it entails quadratic complexity $O(B^2)$ relative to the batch size $B$. More importantly, aligning low-variance local data with high-variance global distributions via RBF can degrade performance due to the homogeneity of local labels (Gretton et al., 2012). Therefore, we adopt the Linear Kernel ($k(x, y) = x^T y$), which simplifies the objective to minimizing the Euclidean distance between mean

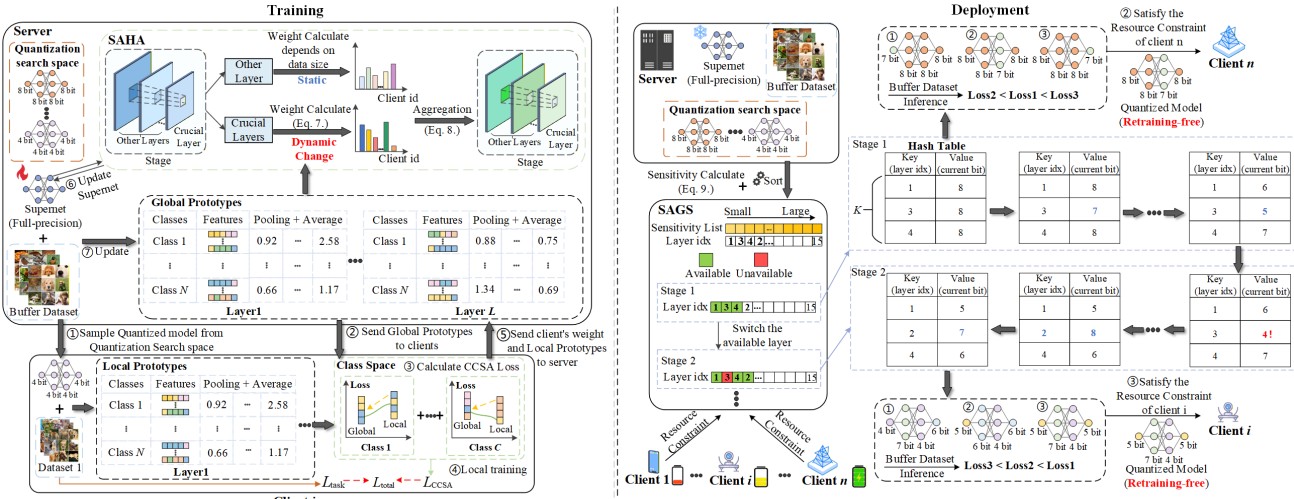

*Figure 3.* Overview of the DFMPQ framework. The approach comprises supernet training utilizing CCSA and SAHA for convergence, followed by deployment where SAGS is applied to search for efficient configurations.

embeddings or centroids. This choice reduces the computational cost to $O(B)$ and facilitates bandwidth-efficient prototype broadcasting. Considering that directly minimizing marginal distributions risks negative transfer under label distribution shifts (Zhang et al., 2013), we implement Class-Conditional Semantic Alignment (CCSA). By calculating the distance between local and global prototypes of the same class, CCSA ensures precise semantic mapping. The regularization term for client $k$ is formulated as:

$$\mathcal{L}_{CCSA}(k) = \sum_{l \in \mathcal{S}} \sum_{c \in \mathcal{C}} \left\| \mu_{k,c}^{(l)} - \mu_{g,c}^{(l)} \right\| \qquad (5)$$

where $\mu_{k,c}^{(l)}$ and $\mu_{g,c}^{(l)}$ denote the local and global feature prototypes for class $c$ at layer $l$. The total loss for client $k$ is a weighted combination of the task loss and this regularization term, governed by the hyperparameter $\lambda$:

$$\mathcal{L}_{\text{total}}(k) = \mathcal{L}_{task}(w_k; \mathcal{D}_k) + \lambda \mathcal{L}_{CCSA}(k) \qquad (6)$$

### 4.2. Semantic-Aware Hybrid Aggregation Strategy

Standard FedAvg calculates a static weight based solely on data volume ($N_k$) and applies it uniformly across all model layers. This approach overlooks the inherent fact that different network layers exhibit varying sensitivities to data heterogeneity. Furthermore, since each client in our framework samples a distinct mixed-precision model in every round, assigning a static and uniform weight based exclusively on data volume while ignoring diverse quantization configurations is suboptimal.To address this, we propose a Hybrid Aggregation Strategy. Specifically, we partition the network into $S$ stages. For the output layer of each stage $s$,

we compute its aggregation weight $w_k^{(s)}$ by incorporating both data volume and semantic confidence as follows:

$$w_k^{(s)} = \frac{N_k \cdot \exp\left(-\frac{\sum_{c \in \mathcal{C}} \|\mu_{k,c}^{(s)} - \mu_{g,c}^{(s)}\|}{\tau}\right)}{\sum_{j=1}^{K} N_j \cdot \exp\left(-\frac{\sum_{c \in \mathcal{C}} \|\mu_{j,c}^{(s)} - \mu_{g,c}^{(s)}\|}{\tau}\right)} \qquad (7)$$

where $\mu_{k,c}^{(s)}$ and $\mu_{g,c}^{(s)}$ denote the local and global prototypes, respectively, and $\tau$ is a temperature hyperparameter. Conversely, all other intermediate layers follow the standard volume-based weighting. Let $W_g^{t+1}$ be the updated global model at round $t + 1$. The aggregation is performed layer by layer. For each layer $l$ belonging to stage $s_m$, its parameters are updated according to the following:

$$W_g^{(l),t+1} = \begin{cases} \sum_{k=1}^{K} w_k^{(m)} W_k^{(l),t+1}, & \text{if } l \text{ is a Crucial Layer} \\ \sum_{k=1}^{K} \frac{N_k}{N} W_k^{(l),t+1}, & \text{otherwise} \end{cases}$$
$$(8)$$

While computing a dynamic aggregation weight for every layer offers finer-grained control, features in deep networks typically evolve in stages, such as residual blocks in ResNet. The feature distributions of intermediate layers often exhibit significant volatility, whereas semantic concepts only become consolidated at the end of each stage. Consequently, a strict layer-wise approach may introduce instability, a phenomenon we have empirically verified.

### 4.3. Sensitivity-Aware Greedy Search

While Evolutionary Algorithms (EA) can effectively explore the quantization configuration search space, they re-

quire maintaining large populations and performing extensive inference evaluations, which incurs prohibitive latency during the deployment phase. To enable real-time model reconfiguration, we propose a Sensitivity-Aware Greedy Search (SAGS) algorithm that leverages gradient priors to efficiently guide the search process.

Specifically, we first compute a static sensitivity score for each layer using the server's cache dataset $\mathcal{D}_{buffer}$ and the full-precision supernet. The sensitivity $\Omega_l$ is defined as the magnitude of the gradient-weight product:

$$\Omega_l = \frac{1}{|\mathbf{W}_l|}\mathbb{E}(x,y) \sim \mathcal{D}_{buffer}\left[\|\frac{\partial \mathcal{L}}{\partial W_l} \odot W_l\|_1\right] \quad (9)$$

Based on $\Omega_l$, we rank all layers in ascending order of sensitivity to construct a priority list $\mathcal{R}$.

Given a target resource constraint $\mathcal{C}_{target}$ (e.g., BitOps), we initialize the search with the maximum configuration $S^{(0)}$. At each step $t$, if the current cost exceeds $\mathcal{C}_{target}$, we select a candidate set $\mathcal{K}_t$ comprising the top-$K$ least sensitive layers from $\mathcal{R}$ that are eligible for downgrading. For each candidate layer $l \in \mathcal{K}_t$, we generate a provisional policy $S_{l\downarrow}^{(t)}$ by reducing its bit-width by one level and evaluate its validation loss via fast inference. The optimal action $a_t$ is determined by minimizing the greedy objective:

$$a_t = \arg\min_{l \in \mathcal{K}_t}; \mathcal{L}_{val}\left(S_{l\downarrow}^{(t)}\right) \quad \text{s.t.} \quad S^{(t+1)} \leftarrow S_{a_t^*\downarrow}^{(t)} \quad (10)$$

This process repeats until the constraint is satisfied. By restricting the search space to the $K$ least sensitive layers, SAGS reduces the search complexity from linear $O(L)$ to constant $O(K)$, achieving significant speedup while preserving model accuracy.

## 5. Theoretical Analysis

### 5.1. Convergence Analysis

First, we theoretically prove that the DFMPQ framework achieves robust convergence. We formally demonstrate that our CCSA regularizer bounds the client drift induced by coupled data and system heterogeneity, while our SAHA strategy approximates the optimal variance-minimizing aggregation, leading to a tighter convergence error bound.

**Lemma 1.** Let the $k$-th client update its model for $T$ local steps with learning rate $\eta$, starting from $\mathbf{w}_g^t$. With $\mu$ being the strong convexity constant introduced by the CCSA regularizer, the expected squared client drift is bounded:

$$\mathbb{E}\left[\|\mathbf{w}_k^{t+1} - \mathbf{w}_g^t\|^2\right] \leq \frac{4(\delta_k^2 + G^2)}{\mu^2}\left(1 + (1-\eta\mu)^T\right) + \frac{2\eta(G^2 + \sigma_{Q,k}^2)}{\mu} \quad (11)$$

where $G^2$ is the upper bound on the gradient norm, and $\sigma_{Q,k}^2$ denotes the variance of the quantization noise. **This result indicates that the CCSA regularizer effectively constrains client drift even in the presence of quantization noise**.

**Proof.** The detailed proof is provided in Supplementary Material C.1

**Lemma 2.** Suppose the estimation error of client $k$ depends on sample size $N_k$ and semantic drift $\delta_k$. Let $\lambda_k^*$ denote the theoretical optimal weight that minimizes the Mean Squared Error (MSE) of the aggregated gradient. The SAHA aggregation weight $w_k^{(s)}$ serves as a first-order approximation to $\lambda_k^*$. Specifically, by establishing the proxy mapping $\delta_k \sim D_k/\tau$, we have:

$$w_k^{(s)} \approx \lambda_k^* + \mathcal{O}(\delta_k^2) \quad (12)$$

**This implies that SAHA asymptotically converges to the variance-minimizing aggregation strategy for crucial layers.**

**Proof.** The detailed proof is provided in Supplementary Material C.2

**Theorem 1. (Convergence of DFMPQ)** Let the global objective function $F(\mathbf{w})$ be $L$-smooth and $\mu$-strongly convex. For a step size satisfying $\eta \leq \frac{1}{L}$, the sequence of iterates generated by DFMPQ satisfies:

$$\mathbb{E}\left[F(\mathbf{w}_T) - F(\mathbf{w}^*)\right] \leq (1-\eta\mu)^T\Delta_0 + \frac{\eta L\sigma_{\text{SAHA}}^2}{2\mu} \quad (13)$$

where $\Delta_0 = F(\mathbf{w}_0) - F(\mathbf{w}^*)$, and $\sigma_{\text{SAHA}}^2$ represents the upper bound on the variance of the aggregated gradient under the SAHA scheme.

**Proof.** By the $L$-smoothness of the objective function, the expected evolution of the global model loss in one iteration is bounded by:

$$\mathbb{E}_t[F(\mathbf{w}_{t+1})] \leq F(\mathbf{w}_t) + \mathbb{E}_t[\langle\nabla F(\mathbf{w}_t), \mathbf{w}_{t+1} - \mathbf{w}_t\rangle] + \frac{L}{2}\mathbb{E}_t[\|\mathbf{w}_{t+1} - \mathbf{w}_t\|^2] \quad (14)$$

Substituting the global update rule $\mathbf{w}_{t+1} = \mathbf{w}_t - \eta \bar{g}_t$, where $\bar{g}_t$ denotes the aggregated gradient:

$$\mathbb{E}_t[F(\mathbf{w}_{t+1})] \leq F(\mathbf{w}_t) - \eta \langle \nabla F(\mathbf{w}_t), \mathbb{E}_t[\bar{g}_t] \rangle \\ + \frac{L\eta^2}{2} \mathbb{E}_t[\|\bar{g}_t\|^2] \quad (15)$$

We define the deviation between the aggregated gradient and the true global gradient as $\mathcal{E}_t = \bar{g}_t - \nabla F(\mathbf{w}_t)$. Using the decomposition $\mathbb{E}_t[\|\bar{g}_t\|^2] = \|\nabla F(\mathbf{w}_t)\|^2 + \mathbb{E}_t[\|\mathcal{E}_t\|^2]$, the bound becomes:

$$\mathbb{E}_t[F(\mathbf{w}_{t+1})] \leq F(\mathbf{w}_t) - \eta\|\nabla F(\mathbf{w}_t)\|^2 \\ + \frac{L\eta^2}{2}\left(\|\nabla F(\mathbf{w}_t)\|^2 + \mathbb{E}_t[\|\mathcal{E}_t\|^2]\right) \quad (16)$$

Rearranging terms and using the condition $\eta \leq \frac{1}{L}$ (which implies $1 - \frac{L\eta}{2} \geq \frac{1}{2}$):

$$\mathbb{E}_t[F(\mathbf{w}_{t+1})] \leq F(\mathbf{w}_t) - \frac{\eta}{2}\|\nabla F(\mathbf{w}_t)\|^2 + \frac{L\eta^2}{2}\underbrace{\mathbb{E}_t[\|\mathcal{E}_t\|^2]}_{\sigma_{\text{agg}}^2} \quad (17)$$

The term $\sigma_{\text{agg}}^2$ captures both the stochastic variance and the systematic drift caused by heterogeneity. According to Lemma 1, the drift-induced deviation is effectively suppressed via the CCSA regularizer. Simultaneously, Lemma 2 establishes that the SAHA weighting scheme asymptotically minimizes the aggregation variance compared to standard averaging, i.e., $\sigma_{\text{agg}}^2 = \sigma_{\text{SAHA}}^2$

Using the $\mu$-strong convexity property $\|\nabla F(\mathbf{w}_t)\|^2 \geq 2\mu(F(\mathbf{w}_t) - F(\mathbf{w}^*))$, we obtain the recurrence relation for the optimality gap $\Delta_t = \mathbb{E}[F(\mathbf{w}_t) - F(\mathbf{w}^*)]$:

$$\Delta_{t+1} \leq (1 - \eta\mu)\Delta_t + \frac{L\eta^2}{2}\sigma_{\text{SAHA}}^2 \quad (18)$$

Unrolling this recursion from $t = 0$ to $T - 1$:

$$\Delta_T \leq (1 - \eta\mu)^T \Delta_0 + \frac{L\eta^2}{2}\sigma_{\text{SAHA}}^2 \sum_{k=0}^{T-1}(1 - \eta\mu)^k \quad (19)$$

By bounding the geometric series sum $\sum_{k=0}^{\infty}(1 - \eta\mu)^k \leq \frac{1}{\eta\mu}$, the final upper bound is derived as:

$$\mathbb{E}[F(\mathbf{w}_T) - F(\mathbf{w}^*)] \leq (1 - \eta\mu)^T \Delta_0 + \frac{L\eta\sigma_{\text{SAHA}}^2}{2\mu} \quad (20)$$

This confirms that the asymptotic error floor is determined by $\sigma_{\text{SAHA}}^2$, establishing the superior convergence properties of the proposed method.

## 5.2. The availability of SAGS

Prior work (Dong et al., 2019) indicates that second-order information yields a more accurate sensitivity measure compared to first-order gradients, which often become unreliable as the model converges to a stationary point. However, computing the Hessian trace incurs a prohibitive computational cost, rendering it infeasible for federated learning scenarios. In this section, we theoretically justify the use of first-order metrics by proving that, in a weight-sharing supernet, the full-precision gradient remains non-zero due to intrinsic task conflicts. Additionally, we prove that this first-order metric acts as a tight upper bound for the quantization-induced loss, establishing $\Omega_l$ as a metric that effectively balances theoretical rigor with computational efficiency

**Theorem 2.** Let $\mathbf{W}^*$ be a stationary point of the scalarized supernet objective $\mathcal{J}(\mathbf{W}) = \sum_{i=1}^{M} p_i \mathcal{L}(\mathbf{W}; \alpha_i)$. Under the assumption that the stationary points of the subnets are distinct (i.e., conflicting optimization landscapes), the gradient of the Full-Precision subnet at $\mathbf{W}^*$ does not vanish:

$$\|\nabla_{\mathbf{W}}\mathcal{L}(\mathbf{W}^*; \alpha_{FP})\| > 0 \quad (21)$$

**Proof.** The detailed proof is provided in Supplementary Material D

**Theorem 3.** Assuming the loss function $\mathcal{L}$ is locally smooth and the quantization error is proportional to the weight magnitude, the increase in loss induced by quantizing layer $l$ is bounded by the $L_1$-norm of the gradient-weight product:

$$|\Delta\mathcal{L}_l| \lesssim \epsilon \cdot \|\nabla_{W_l}\mathcal{L} \odot W_l\|_1 \quad (22)$$

**Proof.** The detailed proof is provided in Supplementary Material E

## 6. Experiments

### 6.1. Setup

We implemented a Federated Learning (FL) simulation environment on a deep learning workstation equipped with NVIDIA A100 (80GB) GPUs. To simulate resource heterogeneity, we configured the participating clients with varying computational capacities. For all experiments, we employed mini-batch Stochastic Gradient Descent (SGD) with an initial learning rate of 0.1, a momentum of 0.9, and a weight decay of 0.0005. The batch size was set to 256. The training process consisted of 50 global rounds, with 5 local epochs per round. The number of clients was set to 10, and the hyperparameter $\lambda$ for the CCSA regularization term was set to 0.1. Following the strategy in (Zhang et al., 2021), we utilized a Dirichlet distribution with a variable concentration parameter $\alpha$ to generate non-IID data partitions. A smaller $\alpha$ indicates a higher degree of non-IID characteristics.

*Table 1.* Test accuracy (%) on CIFAR-10, CIFAR-100, and Tiny-ImageNet under different Dirichlet concentration parameters $\alpha \in \{0.1, 0.5, 1\}$. The number of clients is set to 10, and the hyperparameter $\lambda$ for the CCSA regularization term is 0.1. Columns "Weight" and "Activation" specify the bit-widths used for storing model weights and activation values, respectively. "MP" means mixed-precision quantization. The final column indicates whether the scheme requires retraining in scenarios with dynamic resource constraints.

| Method | CIFAR-10 | | | CIFAR-100 | | | Tiny-ImageNet | | | Weight | Activation | Retraining? |
|---|---|---|---|---|---|---|---|---|---|---|---|---|
| | $\alpha = 0.1$ | 0.5 | 1 | $\alpha = 0.1$ | 0.5 | 1 | $\alpha = 0.1$ | 0.5 | 1 | | | |
| FedAvg | 70.65 | 85.45 | 89.25 | 43.45 | 49.04 | 49.6 | 30.37 | 36.54 | 38.21 | 32 | 32 | ✓ |
| FedProx | 70.89 | 86.60 | 88.81 | 43.78 | 48.49 | 48.54 | 29.99 | 37.14 | 37.43 | 32 | 32 | ✓ |
| UVeQFed | 56.80 | 76.70 | 81.50 | 38.60 | 46.50 | 48.80 | 21.30 | 34.30 | 37.40 | 32 | 32 | ✓ |
| FedWS | 85.94 | 91.0 | 92.47 | 64.14 | 69.31 | 70.14 | 47.05 | 54.23 | 55.26 | 32 | 32 | ✓ |
| FedPAQ | 66.81 | 81.33 | 85.36 | 38.06 | 43.39 | 44.41 | 28.54 | 33.58 | 34.68 | 32 | 32 | ✓ |
| FedHQ+ | 66.89 | 81.37 | 85.32 | 38.22 | 42.91 | 44.77 | 28.59 | 34.24 | 35.55 | 32 | 32 | ✓ |
| AQFL | 44.30 | 58.00 | 62.10 | 23.80 | 32.90 | 36.10 | 17.10 | 23.50 | 25.30 | MP | 4 | ✓ |
| FedMPQ | 49.10 | 67.10 | 69.30 | 31.70 | 41.10 | 43.60 | 20.30 | 27.00 | 28.20 | MP | 4 | ✓ |
| DFMPQ (ours) | 78.86 | 80.24 | 89.27 | 62.22 | 64.15 | 66.62 | 64.95 | 67.29 | 68.05 | 3MP | 3MP | ✗ |

*Table 2.* Test accuracy (%) on CIFAR-10 and CIFAR-100 for various numbers of clients $N \in \{10, 20, 40\}$ with a Dirichlet concentration parameter $\alpha = 0.5$. In our method, both weights and activations are quantized using mixed-precision with an average bit-width of 3 bits.

| Method | CIFAR-10 | | | CIFAR-100 | | |
|---|---|---|---|---|---|---|
| | $N = 10$ | 20 | 40 | $N = 10$ | 20 | 40 |
| FP32 | 77.5 | 68.1 | 64.5 | 47.0 | 40.1 | 35.0 |
| FedPAQ | 77.0 | 61.5 | 59.9 | 46.8 | 38.9 | 33.1 |
| UVeQFed | 76.7 | 66.4 | 63.2 | 46.5 | 39.5 | 34.1 |
| FPQ8 | 68.4 | 56.3 | 48.4 | 41.3 | 36.2 | 31.9 |
| AQFL | 58.0 | 49.7 | 37.3 | 32.9 | 20.4 | 13.9 |
| FedMPQ | 67.1 | 56.8 | 45.4 | 41.1 | 26.1 | 19.9 |
| DFMPQ | 80.24 | 79.14 | 75.42 | 64.15 | 57.61 | 36.64 |

## 6.2. Effects of Data Heterogeneity

We evaluate the impact of data heterogeneity by setting the Dirichlet concentration parameter $\alpha \in \{0.1, 0.5, 1.0\}$. Results in Table 1 show performance improves as heterogeneity decreases. Our method matches the full-precision FedWS(Kim et al., 2025) using only 3 bits and significantly outperforms fixed-precision baselines (Fed-PAQ(Reisizadeh et al., 2020), FedHQ+(Chen et al., 2021a), AQFL(Abdelmoniem & Canini, 2021)). Notably, it surpasses the mixed-precision FedMPQ(Chen & Vikalo, 2024) by over 20% across CIFAR-10 and CIFAR-100(Krizhevsky et al., 2009) and Tiny-ImageNet(Le & Yang, 2015). This advantage stems from our framework's efficacy and its focus on dynamic bit-width optimization, unlike FedMPQ's focus on static hardware constraints.

## 6.3. Different Local Epochs

To investigate the impact of local training epochs on system performance, we evaluated settings across the set $\{1, 5, 10, 15, 20\}$. As illustrated in Figure 4, existing methods lack a consistent positive correlation between performance and the number of local epochs. Conversely, our proposed method exhibits a robust positive trend: although performance gains show diminishing returns as epochs increase, the trajectory remains upward. Moreover, our approach outperforms the state-of-the-art mixed-precision method, FedMPQ, achieving higher accuracy with lower resource consumption.

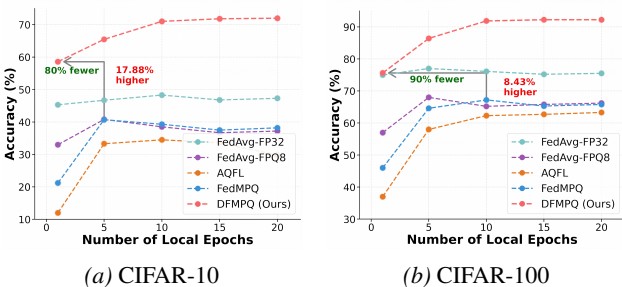

*(a) CIFAR-10*      *(b) CIFAR-100*

*Figure 4.* Top-1 accuracy of ResNet18 on CIFAR-10 and CIFAR-100 with varying local training epochs.

## 6.4. Different Number of Clients

To evaluate the scalability of DFMPQ under varying system sizes, we conducted experiments on the CIFAR-10 and CIFAR-100 datasets with 10, 20, and 40 participating clients. The data heterogeneity was fixed with a Dirichlet concentration parameter $\alpha = 0.5$. Table 2 indicates that while the performance of all compared methods degrades as the system scale expands, DFMPQ consistently achieves superior accuracy. Notably, this is accomplished using an average precision of only 3 bits for both weights and activations.

## 6.5. Architecture Generalization

To verify the architectural generalization capability of our DFMPQ framework, we extended our evaluation to in-

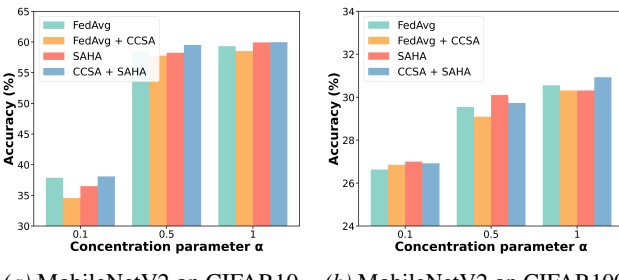

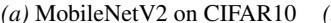

*(a)* MobileNetV2 on CIFAR10    *(b)* MobileNetV2 on CIFAR100

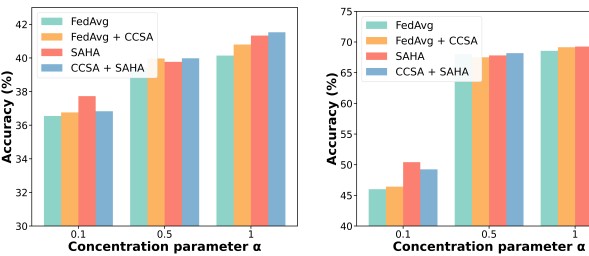

*(c)* EfficientNet-Lite on CIFAR-10    *(d)* EfficientNet-Lite on CIFAR-100

*Figure 5.* Performance evaluation of DFMPQ across diverse edge-compatible architectures (MobileNetV2 and EfficientNet-Lite0) on CIFAR-10 and CIFAR-100 under varying Non-IID settings.

clude diverse modern network architectures, specifically the lightweight MobileNetV2 and EfficientNet-Lite0. As prevalent backbones in resource-constrained edge computing, these models serve as representative benchmarks for our approach. All experiments were conducted on the CIFAR-10 and CIFAR-100 datasets under non-IID settings ($\alpha = \{0.1, 0.5, 1.0\}$), with hyperparameters consistent with the primary experiments.

The empirical results, illustrated in Figure 5, substantiate that the core mechanisms of our framework, specifically CCSA and SAHA, are inherently model-agnostic. By operating within the feature space, these components effectively mitigate the challenges posed by non-IID data and heterogeneous quantization noise. This independence from specific convolutional structures or network depths confirms the robustness and practical applicability of DFMPQ across a broad spectrum of edge-compatible models.

### 6.6. Ablation Study

**Hyperparameter $\lambda$.** To investigate the impact and robustness of our CCSA module, we conducted a sensitivity analysis on its balancing hyperparameter, $\lambda$. Using a ResNet18 backbone, we evaluated the accuracy of a mixed-precision model (average 3-bit for weights and activations) across the CIFAR-10, CIFAR-100, and Tiny-ImageNet datasets. All datasets were partitioned into non-IID distributions with $\alpha = 0.5$, while $\lambda$ was varied across the set $\{0, 0.01, 0.05, 0.1, 0.2\}$. The results, illustrated in Figure 6,

exhibit consistent trends across all three datasets.

First, the baseline case with $\lambda = 0$ (i.e., abating the CCSA module) yields the lowest accuracy. This consistent underperformance underscores the critical role of CCSA in mitigating data heterogeneity. Second, accuracy improves as $\lambda$ increases up to 0.1, indicating that a moderate regularization force effectively guides local models towards a global semantic consensus. Finally, when $\lambda$ becomes excessive ($\lambda > 0.1$), performance begins to decline. This is attributed to over-regularization, where overly strong alignment constraints hinder the model's ability to fit local data characteristics. Consequently, we set $\lambda = 0.1$ for all subsequent experiments to ensure an optimal trade-off between semantic alignment and local adaptation.

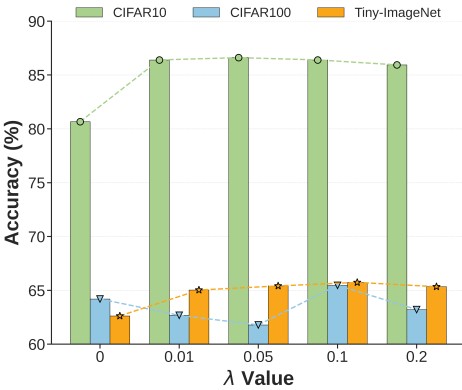

*Figure 6.* Impact of the CCSA regularization coefficient $\lambda$ on model test accuracy across different datasets

**Aggregation Strategy.** To validate the efficacy of our Semantic-Aware Hybrid Aggregation (SAHA) design, we benchmark our proposed strategy against several alternative schemes. Experiments are conducted using ResNet-18 under a non-IID setting ($\alpha = 0.5$), with 50 global rounds, 5 local epochs, and $\lambda = 0.1$. The compared strategies are defined as follows:

- **FedAvg:** The standard baseline that applies uniform, sample-volume-based aggregation weights ($\lambda_k \propto N_k$) across all layers.

- **Layer-wise:** A fine-grained approach where a unique semantic-aware weight is computed and applied individually to every layer.

- **Stage-wise:** A block-level approach where the semantic weight derived from a stage's output layer is applied to all internal layers within that stage.

- **Hybrid (Ours):** Our proposed strategy, which applies semantic-aware weights exclusively to stage output layers, while reverting to standard FedAvg weighting for intermediate layers.

As presented in Table 3, despite its fine-grained nature, the **Layer-wise** strategy counter-intuitively yields the worst performance, often underperforming the FedAvg baseline. This failure stems from its reliance on volatile intermediate features. Feature maps in intermediate layers are typically semantically immature and highly susceptible to cumulative quantization noise. Consequently, semantic confidence scores derived from these low signal-to-noise ratio (SNR) features are unreliable, resulting in erratic aggregation weights that destabilize training.

In contrast, both **Stage-wise** and **Hybrid** strategies significantly outperform **Layer-wise**. This superiority arises because both methods derive semantic confidence scores exclusively from stage output representations. At these topological checkpoints, features are semantically consolidated and resilient to noise, providing a robust indicator of model quality. Notably, on the complex CIFAR-100 dataset, our Hybrid strategy achieves 70.88% accuracy, surpassing FedAvg (70.69%).

Finally, while both strategies are competitive, our **Hybrid** approach offers a more robust design principle. Since bit-width configurations within a residual stage vary dynamically, the Stage-wise strategy imposes a single semantic weight on the entire block, which may lead to over-regularization of intermediate layers. Our Hybrid strategy resolves this by applying strict semantic filtering only at critical stage outputs, while allowing intermediate layers to follow the stable, volume-based FedAvg protocol. This design strikes an optimal balance between targeted quality control and training stability, confirming the rationale behind selecting the Hybrid strategy for the DFMPQ framework.

*Table 3.* Performance comparison of aggregation strategies with different semantic granularities.

|  | FedAvg | Layer | Stage | Hybrid |
|---|---|---|---|---|
| CIFAR10 | 90.42 | 89.74 | **90.92** | 90.71 |
| CIFAR100 | 70.69 | 70.19 | 70.50 | **70.88** |
| Tiny-ImageNet | 66.23 | 63.70 | 66.26 | **66.65** |

**We provide robustness experiments regarding the server buffer dataset $\mathcal{D}_{buffer}$ in Supplementary Material A, and visualizations of feature drift using DFMPQ in Supplementary Material B.**

## 7. Related Works

**Mixed-Precision Quantization (MPQ)** assigns heterogeneous bit-widths to balance accuracy and efficiency (Cai & Vasconcelos, 2020; Tang et al., 2022), creating a challenge within an exponential solution space (Tang et al., 2024). *Search-based methods* (e.g., DNAS (Wu et al., 2018),

SEAM (Tang et al., 2023)) formulate allocation as a differentiable search. To address gradient instability, SDQ (Huang et al., 2022) and NIPQ (Shin et al., 2023) employ stochastic relaxation and pseudo-quantization noise, respectively, while Entropy-Driven MPQ (Sun et al., 2022) expands to joint architecture optimization. In contrast, *metric-based methods* assess layer importance via various metrics to determine allocation. For instance, HAWQ (Dong et al., 2019) and MPQCO (Chen et al., 2021b) utilize Hessian-based information. To avoid the computation overhead of Hessians, OMPQ (Ma et al., 2023) and GMPQ-TE (Li et al., 2026) leverage orthogonality and topological entropy via fast linear programming, respectively.

**Quantization for Federated Learning.** While Federated Learning (FL) enables collaborative training with privacy preservation (McMahan et al., 2017), it incurs significant communication overhead. Early works, such as FedPAQ (Reisizadeh et al., 2020) and FedCOMGATE (Haddadpour et al., 2021), addressed this via weight or gradient quantization. More recently, research has shifted towards finer granularity; for instance, FedFQ (Li et al., 2024) introduces parameter-level adaptive quantization to balance high compression ratios with convergence performance. To tackle device heterogeneity in edge scenarios, methods like BHFL (Yoon et al., 2022) and others (Chen et al., 2021a; Elkordy & Avestimehr, 2022) have explored aggregating updates with varying quantization levels. For resource-constrained IoT devices, FedX (Lai et al., 2025) combines adaptive model decomposition with quantization to accommodate diverse hardware capabilities. However, despite these advancements, most approaches still rely on fixed-precision or slowly adapting schedules, which restricts the accuracy-efficiency trade-off. Although FedMPQ (Chen & Vikalo, 2024) introduced MPQ into heterogeneous FL, it necessitates time-consuming retraining when device resource conditions change, rendering it impractical for highly dynamic environments.

## 8. Conclusions

In this paper, we presented DFMPQ to address the challenges of efficient model deployment in dynamic edge environments. By leveraging federated learning to train a supernet supporting diverse bit-widths, our approach enables the generation of retraining-free quantized networks. To mitigate bit-width interference during training and feature drift exacerbated by quantization noise, we introduced CCSA for global class-conditional semantic alignment and SAHA for semantic quality-aware aggregation. Furthermore, we developed the SAGS strategy, which exploits gradient priors to efficiently search for optimal networks. Extensive experiments and theoretical analysis validate the effectiveness of the proposed framework.

## Acknowledgements

This work is supported by National Natural Science Foundation of China under Grant 62472079

## Impact Statement

This paper presents an approach, called DFMPQ, which fills the gap in the efficient deployment of quantized networks in dynamic edge environments. There are many potential societal consequences of our work, none which we feel must be specifically highlighted here.

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

## Overview of the Supplementary Material

The main contents of this Supplementary Material are as follows:

- Section A Robustness Analysis of $\mathcal{D}_{buffer}$

- Section B presents visualizations of the feature drift using DFMPQ, serving as a comparison to Figure 2.

- Section C provides the relevant parts for Theorem 1. It first presents the preliminary assumptions for the subsequent proof of Theorem 1, and provides the complete proofs of the lemmas in C.1 and C.2, respectively.

- Section D Proof of Theorem 2.

- Section E Proof of Theorem 3.

## A. Robustness Analysis of $\mathcal{D}_{buffer}$

To evaluate the impact of the per-class buffer size and data distribution on the SAGS module, we conducted two sets of comparative experiments. In the first set, we varied the per-class sample count in $\mathcal{D}_{buffer}$ from 5 to 50, and the corresponding results are reported in Table 4. In the second set, we varied the distributional heterogeneity by adjusting the Dirichlet parameter $\alpha \in \{0.1, 0.5, 1.0, 100\}$, with the performance summarized in Table 5.

Across all these configurations, accuracy fluctuations remained within a negligible range. This stability provides strong empirical evidence that Eq. (9) captures an intrinsic property of the model architecture, rather than merely fitting the distributional characteristics of a specific buffer dataset.

*Table 4.* Robustness to Per-Class Buffer Size in $\mathcal{D}_{buffer}$

| Samples/class | CIFAR-10 | CIFAR-100 | Tiny-ImageNet |
|:---:|:---:|:---:|:---:|
| 5 | 89.46 | 65.53 | 66.95 |
| 10 | 89.36 | 65.46 | 66.94 |
| 20 (default) | 89.85 | 65.84 | 66.65 |
| 50 | 89.08 | 65.16 | 66.61 |

*Table 5.* Robustness to heterogeneity in $\mathcal{D}_{buffer}$

| Buffer $\alpha$ | CIFAR-10 | CIFAR-100 | Tiny-ImageNet |
|:---:|:---:|:---:|:---:|
| 0.1 | 89.36 | 65.42 | 66.95 |
| 0.5 | 88.76 | 65.46 | 66.54 |
| 1 | 90.05 | 65.27 | 66.27 |
| 100 | 89.34 | 65.16 | 66.4 |

## B. Visualization of Feature Drift under DFMPQ

To validate the effectiveness of DFMPQ in mitigating the feature shift between client-side mixed-precision models and the server-side global model, we conducted experiments using the same setup as in Figure 2. The results are illustrated in Figure 7, where dashed lines connect the feature representations of the same sample. A comparison with Figure 2b demonstrates that our method significantly mitigates feature drift.

## C. Proof for Theorem 1

Our convergence analysis is based on the following standard assumptions, which are common in the federated optimization literature.

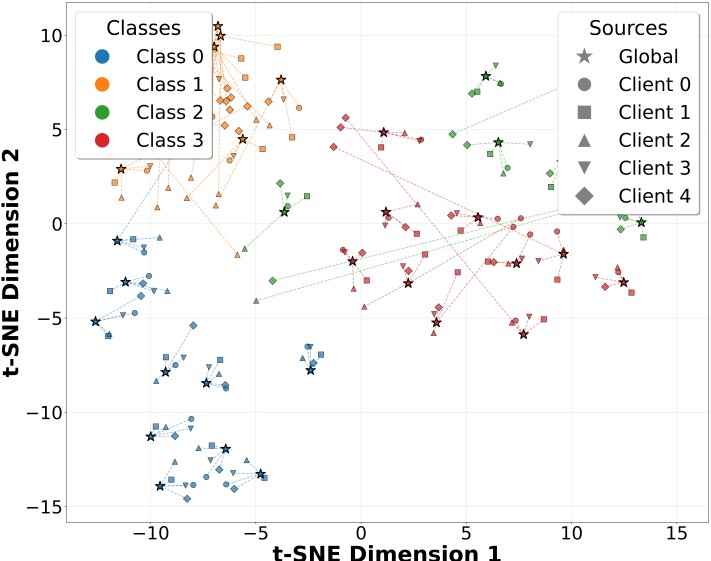

*Figure 7.* Feature Shift under DFMPQ

**Assumption C.1** (**L-smoothness**). The global loss function $F(\mathbf{w})$ and each local loss function $F_k(\mathbf{w})$ are $L$-smooth. This means their gradients are Lipschitz continuous with constant $L > 0$:

$$\|\nabla F(\mathbf{w}_1) - \nabla F(\mathbf{w}_2)\| \le L \cdot \|\mathbf{w}_1 - \mathbf{w}_2\|$$

**Assumption C.2** ($\mu$**-strong convexity**). The global loss function $F(\mathbf{w})$ is $\mu$-strongly convex with constant $\mu > 0$. This ensures a unique global minimum $\mathbf{w}^*$:

$$F(\mathbf{w}_1) \ge F(\mathbf{w}_2) + \langle \nabla F(\mathbf{w}_2), \mathbf{w}_1 - \mathbf{w}_2 \rangle + \frac{\mu}{2} \cdot \|\mathbf{w}_1 - \mathbf{w}_2\|^2$$

**Assumption C.3** (**Bounded Gradients and Heterogeneity**). The training process exhibits bounded noise and heterogeneity:

(i) **Bounded Gradient Norm**: The expected squared norm of the stochastic local gradients is bounded by a constant $G^2$:

$$\mathbb{E}\left[\|\nabla F_k(\mathbf{w})\|^2\right] \le G^2$$

(ii) **Bounded Data Heterogeneity**: The dissimilarity between local and global true gradients is bounded:

$$\mathbb{E}\left[\|\nabla F_k(\mathbf{w}) - \nabla F(\mathbf{w})\|^2\right] \le \delta_k^2$$

(iii) **Bounded System Heterogeneity**: The variance of the gradient due to quantization is bounded:

$$\mathbb{E}\left[\|\hat{g}_k(\mathbf{w}) - g_k(\mathbf{w})\|^2\right] \le \sigma_{Q,k}^2$$

where $\hat{g}_k$ is the quantized gradient and $g_k$ is the true local gradient.

**Assumption C.4** (**Properties of Regularizer**). The CCSA regularization term $\mathcal{L}_{\text{CCSA}}(\mathbf{w}) = \sum_{l,c} \left\|\mu_{k,c}^{(l)}(\mathbf{w}) - \mu_{g,c}^{(l)}\right\|^2$, is $\mu$-strongly convex and $L$-smooth.

In the following section, we provide the proofs for Lemma 1 and Lemma 2.

### C.1. Proof for Lemma 1

**Lemma 1. (Bounded Client Drift with Class-Conditional Semantic Alignment).** Let the $k$-th client update its model for $T$ local steps with learning rate $\eta$ starting from $\mathbf{w}_g^t$. Under Assumptions A.1–A.4, and letting $\mu$ be the strong convexity constant introduced by the CCSA regularizer, the expected squared client drift is bounded: $\mathbb{E}\left[\left\|\mathbf{w}_k^{t+1} - \mathbf{w}_g^t\right\|^2\right] \le$
$\frac{4(\delta_k^2+G^2)}{\mu^2}\left(1 + (1-\eta\mu)^T\right) + \frac{2\eta(G^2+\sigma_{Q,k}^2)}{\mu}$

*Proof.* Let the local objective function be $\mathcal{F}_k(\mathbf{w}) = \mathcal{L}_{\text{task},k}(\mathbf{w}) + \lambda\mathcal{L}_{\text{CCSA},k}(\mathbf{w})$. Based on Assumptions A.2 and A.4, $\mathcal{F}_k(\mathbf{w})$ is $\mu$-strongly convex. Let $\mathbf{w}_k^* = \arg\min_{\mathbf{w}} \mathcal{F}_k(\mathbf{w})$ be the local optimum.

The total client drift can be decomposed via the triangle inequality as follows:

$$\mathbb{E}\left[\left\|\mathbf{w}_k^{t+1} - \mathbf{w}_g^t\right\|^2\right] \leq 2\mathbb{E}\left[\left\|\mathbf{w}_k^{t+1} - \mathbf{w}_k^*\right\|^2\right] + 2\left\|\mathbf{w}_k^* - \mathbf{w}_g^t\right\|^2 \tag{23}$$

First, we derive an upper bound for $\mathbb{E}\left[\left\|\mathbf{w}_k^{t+1} - \mathbf{w}_k^*\right\|^2\right]$. The local update at step $\tau$ uses a stochastic gradient $\hat{g}_k$ corrupted by quantization noise. Expanding the distance to the local optimum for one step:

$$\begin{aligned}
\mathbb{E}\left[\left\|\mathbf{w}_k^{t,\tau+1} - \mathbf{w}_k^*\right\|^2\right] &= \mathbb{E}\left[\left\|\mathbf{w}_k^{t,\tau} - \eta\hat{g}_k - \mathbf{w}_k^*\right\|^2\right] \\
&\leq (1-\eta\mu)\mathbb{E}\left[\left\|\mathbf{w}_k^{t,\tau} - \mathbf{w}_k^*\right\|^2\right] + \eta^2(G^2 + \sigma_{\text{Q},k}^2)
\end{aligned} \tag{24}$$

Recursively applying this contraction for $T$ steps and summing the resulting geometric series yields the bound on the local convergence:

$$\mathbb{E}\left[\left\|\mathbf{w}_k^{t+1} - \mathbf{w}_k^*\right\|^2\right] \leq (1-\eta\mu)^T\left\|\mathbf{w}_g^t - \mathbf{w}_k^*\right\|^2 + \frac{\eta(G^2 + \sigma_{\text{Q},k}^2)}{\mu} \tag{25}$$

Next, we derive an upper bound for $\|\mathbf{w}_g^t - \mathbf{w}_k^*\|^2$. Using the property of $\mu$-strong convexity, $\|\mathbf{w} - \mathbf{w}^*\|^2 \leq \frac{1}{\mu^2}\|\nabla\mathcal{F}_k(\mathbf{w})\|^2$. Applying this at $\mathbf{w}_g^t$ and using the triangle inequality:

$$\begin{aligned}
\left\|\mathbf{w}_g^t - \mathbf{w}_k^*\right\|^2 &\leq \frac{1}{\mu^2}\left\|\nabla\mathcal{F}_k(\mathbf{w}_g^t) - \nabla F(\mathbf{w}_g^t) + \nabla F(\mathbf{w}_g^t)\right\|^2 \\
&\leq \frac{2}{\mu^2}\left(\left\|\nabla\mathcal{F}_k(\mathbf{w}_g^t) - \nabla F(\mathbf{w}_g^t)\right\|^2 + \left\|\nabla F(\mathbf{w}_g^t)\right\|^2\right)
\end{aligned} \tag{26}$$

By invoking Assumption A.3, we obtain a strict upper bound:

$$\left\|\mathbf{w}_g^t - \mathbf{w}_k^*\right\|^2 \leq \frac{2(\delta_k^2 + G^2)}{\mu^2} \tag{27}$$

Based on the derivation above, we obtain:

$$\mathbb{E}\left[\left\|\mathbf{w}_k^{t+1} - \mathbf{w}_g^t\right\|^2\right] \leq \frac{4(\delta_k^2 + G^2)}{\mu^2}\left(1 + (1-\eta\mu)^T\right) + \frac{2\eta(G^2 + \sigma_{\text{Q},k}^2)}{\mu} \tag{28}$$

This confirms that the total client drift is bounded by the data heterogeneity $\delta_k$ and quantization noise $\sigma_{\text{Q},k}$, with the CCSA regularizer effectively constraining the drift. □

### C.2. Proof for Lemma 2

**Lemma 2.** The SAHA aggregation weight $w_k^{(s)}$ asymptotically approximates the optimal variance-minimizing weighting strategy.

*Proof.* The objective of the aggregation phase is to minimize the expected error of the global model update. In a federated setting with non-IID data, the reliability of a client's update is governed by two principal factors:

- **Statistical Confidence** ($N_k$)**:** According to the Law of Large Numbers, the variance of the local stochastic gradient estimator scales inversely with the sample size (Var $\propto 1/N_k$). Thus, standard FedAvg assigns weights proportional to $N_k$ to minimize global variance.

- **Systematic Deviation ($\delta_k$):** Heterogeneity introduces a systematic bias (drift) $\delta_k$ relative to the global distribution. The optimal strategy requires penalizing clients with large drifts.

Combining these factors, the theoretical optimal weight $\lambda_k^*$ that minimizes the Mean Squared Error (MSE) is proportional to the sample size and inversely proportional to the drift magnitude:

$$\lambda_k^* \propto \frac{N_k}{1 + \delta_k} \tag{29}$$

where $\delta_k$ represents the relative magnitude of the semantic shift.

Our proposed SAHA weighting scheme is given by:

$$w_k^{(s)} \propto N_k \cdot \exp\left(-\frac{D_k}{\tau}\right) \tag{30}$$

where the semantic distance $D_k$ serves as a measurable proxy for the drift $\delta_k$.

To demonstrate the optimality of $w_k^{(s)}$, we examine the behavior of both weighting functions for small drift values using the First-order Taylor Expansion:

- Expanding the optimal inverse term around $\delta_k \approx 0$:

$$\frac{1}{1 + \delta_k} = 1 - \delta_k + \mathcal{O}(\delta_k^2) \tag{31}$$

- Expanding the SAHA exponential term around $D_k/\tau \approx 0$:

$$\exp\left(-\frac{D_k}{\tau}\right) = 1 - \frac{D_k}{\tau} + \mathcal{O}\left(\left(\frac{D_k}{\tau}\right)^2\right) \tag{32}$$

By establishing the proxy mapping $\delta_k \sim D_k/\tau$, we confirm that the SAHA weight $w_k^{(s)}$ is mathematically equivalent to the theoretical optimal weight $\lambda_k^*$ up to the first order:

$$w_k^{(s)} \approx N_k \left(1 - \frac{D_k}{\tau}\right) \approx \lambda_k^* \tag{33}$$

This confirms that SAHA effectively approximates the optimal variance-reduction strategy. $\qquad\square$

## D. Proof for Theorem 2

Let $\mathcal{L}(\mathbf{W}; \alpha_i)$ denote the loss function of the $i$-th subnet configuration $\alpha_i$, where $i \in \{1, \ldots, M\}$ and $M$ is the total number of possible subnets. The goal of supernet training is to find a single optimal weight vector $\mathbf{W}^*$ that minimizes the losses of all $M$ subnets concurrently. Formally, this is a Multi-Objective Optimization (MOO) problem:

$$\min_{\mathbf{W}} \mathbf{L}(\mathbf{W}) = [\mathcal{L}(\mathbf{W}; \alpha_1), \ldots, \mathcal{L}(\mathbf{W}; \alpha_M)]^\top \tag{34}$$

where $\mathbf{L}(\mathbf{W}) : \mathbb{R}^d \to \mathbb{R}^M$ is the vector-valued objective function.

Crucially, these objectives are often conflicting. For example, low-bit subnets require weights that are robust to aggressive quantization noise, whereas high-bit subnets require weights that capture fine-grained feature details. Due to these conflicting requirements, there typically does not exist a single $\mathbf{W}$ that minimizes all $\mathcal{L}(\mathbf{W}; \alpha_i)$ simultaneously. Instead, the solution lies on the Pareto Frontier, representing a trade-off among different subnets.

This MOO problem is solved via Linear Scalarization, which aggregates the vector objective into a single scalar loss using sampling probabilities $p_i$:

$$\min_{\mathbf{W}} \mathcal{J}(\mathbf{W}) = \sum_{i=1}^{M} p_i \mathcal{L}(\mathbf{W}; \alpha_i) \tag{35}$$

where $p_i$ represents the probability of sampling subnet $\alpha_i$ during training. The standard supernet training procedure is mathematically equivalent to minimizing this scalarized objective $\mathcal{J}(\mathbf{W})$.

Since $\mathbf{W}^*$ is a stationary point of the scalarized objective $\mathcal{J}(\mathbf{W})$, it must satisfy the first-order optimality condition where the gradient of the aggregate loss vanishes:

$$\nabla \mathcal{J}(\mathbf{W}^*) = \sum_{i=1}^{M} p_i \nabla \mathcal{L}(\mathbf{W}^*; \alpha_i) = \mathbf{0} \tag{36}$$

We decompose this summation into the term for the Full-Precision subnet ($\alpha_{FP}$) and the weighted sum of gradients for all other quantized subnets ($\alpha_j \in \mathcal{A} \setminus \{\alpha_{FP}\}$):

$$p_{FP} \nabla \mathcal{L}(\mathbf{W}^*; \alpha_{FP}) + \sum_{j \neq FP} p_j \nabla \mathcal{L}(\mathbf{W}^*; \alpha_j) = \mathbf{0} \tag{37}$$

Suppose, for the sake of contradiction, that the gradient of the Full-Precision subnet vanishes at this point, i.e., $\nabla \mathcal{L}(\mathbf{W}^*; \alpha_{FP}) = \mathbf{0}$. Substituting this into Eq. (36), we obtain:

$$\sum_{j \neq FP} p_j \nabla \mathcal{L}(\mathbf{W}^*; \alpha_j) = \mathbf{0} \tag{38}$$

Eq. (37) implies that $\mathbf{W}^*$ is also a stationary point for the aggregated objective of the remaining low-bit subnets.

However, due to the regularization effect of quantization noise, the optimal weight distribution for low-bit subnets (which must be robust to noise) differs fundamentally from that of the full-precision subnet. As stated in the assumption of Theorem 2, the sets of stationary points for the FP task and the aggregated low-bit tasks are disjoint:

$$\{\mathbf{W} \mid \nabla \mathcal{L}(\mathbf{W}; \alpha_{FP}) = \mathbf{0}\} \cap \left\{ \mathbf{W} \mid \sum_{j \neq FP} p_j \nabla \mathcal{L}(\mathbf{W}; \alpha_j) = \mathbf{0} \right\} = \emptyset \tag{39}$$

Thus, $\mathbf{W}^*$ cannot simultaneously satisfy $\nabla \mathcal{L}_{FP}(\mathbf{W}^*) = \mathbf{0}$ and Eq. (37). This contradiction invalidates the assumption that $\nabla \mathcal{L}(\mathbf{W}^*; \alpha_{FP}) = \mathbf{0}$, thereby proving that $\|\nabla \mathcal{L}(\mathbf{W}^*; \alpha_{FP})\| > 0$.

## E. Proof for Theorem 3

Let $\Delta \mathbf{W}_l$ denote the perturbation introduced by quantizing the weights $\mathbf{W}_l$ of layer $l$. The change in the loss function can be approximated using the first-order Taylor expansion around the current weights $\mathbf{W}$:

$$\Delta \mathcal{L} = \mathcal{L}(\mathbf{W} + \Delta \mathbf{W}_l) - \mathcal{L}(\mathbf{W}) \approx \langle \nabla_{\mathbf{W}_l} \mathcal{L}, \Delta \mathbf{W}_l \rangle \tag{40}$$

where higher-order terms (e.g., involving the Hessian) are neglected under the assumption that the quantization perturbation $\Delta \mathbf{W}_l$ is sufficiently small (i.e., $\epsilon \ll 1$).

For standard linear quantization schemes, the quantization noise is bounded and typically proportional to the magnitude of the weights (relative error). Specifically, for a given bit-width, the element-wise error can be modeled as:

$$|\Delta w_{l,i}| \leq \epsilon \cdot |w_{l,i}| \tag{41}$$

where $\epsilon$ is a scaling factor depending on the quantization precision, and $i$ indexes the elements of the weight tensor.

Applying the Triangle Inequality and the properties of the inner product, the upper bound of the loss perturbation is:

$$|\Delta\mathcal{L}| \approx \left|\sum_i \frac{\partial\mathcal{L}}{\partial w_{l,i}} \cdot \Delta w_{l,i}\right| \leq \sum_i \left|\frac{\partial\mathcal{L}}{\partial w_{l,i}}\right| \cdot |\Delta w_{l,i}|$$

$$\leq \sum_i \left|\frac{\partial\mathcal{L}}{\partial w_{l,i}}\right| \cdot (\epsilon|w_{l,i}|) \tag{42}$$

$$= \epsilon \sum_i \left|\frac{\partial\mathcal{L}}{\partial w_{l,i}} \cdot w_{l,i}\right| = \epsilon \cdot \|\nabla_{\mathbf{W}_l}\mathcal{L} \odot \mathbf{W}_l\|_1$$

