# OpenReview forum: "No Retraining at Edge: Efficient Resource-Aware Mixed-Precision Quantization via Federated Supernet Learning"
_ICML.cc/2026/Conference — ICML 2026 regular_

### Official Review · Reviewer_AfS2 · 2026-03-09

**Soundness:** 2
**Presentation:** 3
**Significance:** 2
**Originality:** 3
**Overall Recommendation:** 4
**Confidence:** 4

**Summary:**

This paper primarily addresses the retraining problem in federated learning models arising from the complex and variable edge environment. The power and computing capacity of edge devices may change, and existing methods require retraining, which incurs high training costs. Therefore, to alleviate the optimization interference among heterogeneous bit-widths and the feature drift caused by the coupling of quantization noise with non-IID data, the authors propose the DFMPQ method, a dynamic federated mixed-precision quantization framework, which does not require retraining. On the CIFAR-10, CIFAR-100, and Tiny-ImageNet datasets, DFMPQ achieves results comparable to those of full-precision models with low bit widths without retraining.

**Compliance With Llm Reviewing Policy:**

Affirmed.

**Final Justification:**

The authors have addressed most of my concerns, thus I have increased the final rating to 4.

**Key Questions For Authors:**

Here are my key questions.

(1) In Table 1, DFMPQ is compared with multiple fixed-precision and mixed-precision benchmarks. However, these benchmark models employ specific methods to maintain their effectiveness. In your experiment, do these benchmark models include their own specific stability techniques or merely maintain the basic implementation of the benchmark models?

(2) The SAHA algorithm, based on semantics, calculates weights based on semantic confidence during the critical network stage. If a deeper network is used, will the criteria for dividing the critical network remain the same? Will the calculation of stage weights have a significant impact? I am particularly interested in this.

(3) Given the rapid development in the field of edge computing, is your proposed method still applicable to large language models such as Transformers and larger datasets? The current experiments are limited to relatively small datasets and neural networks. How are the generalization and applicability of your method in other aspects?

**Limitations:**

As discussed in the weakness part, I suggest that the authors revise the paper carefully in the following aspects.

(1) The authors could explore the limitations of the research work and promote the development of federated learning.

(2) The current experimental evaluation is limited to being effective in the computer vision tasks of CNN, but currently, federated learning is increasingly being applied in more types of neural networks and NLP scenarios. It has not been explored whether the proposed method has broad applicability or if there are any limitations.

(3) Although the paper has significant implications for the deployment on edge devices, it is unclear whether experiments have been conducted on actual edge devices to demonstrate the applicability of the method.

**Strengths And Weaknesses:**

**Strengths**
The authors aim to address the problems caused by existing quantization noise and non-independent and identically distributed (non-IID) data, such as feature drift during training of the super-network, optimization conflicts in different bit-width network configurations during training, and the high re-training costs caused by resource changes.

**Weaknesses**
I'm not an expert in this field, so please correct me if any parts of my understanding are wrong. There are a few weaknesses that stop me from giving a higher rating. However, I'm still willing to read the author's responses to decide the final ratings. My major concerns are as follows.

(1) My primary concern is about the evaluation, which is not conducted on actual edge hardware. The authors have not presented a detailed analysis of the real-time latency, energy consumption, and computing power limitations under fluctuating hardware conditions. This makes the research motivation of its method seem insufficient.

(2)  I believe that in the context of the rapid development of edge computing, the experiments in this paper are not convincing enough, and the wide applicability of the method is questionable. Specifically, the method in this article is based on smaller-scale datasets, such as CIFAR-10/100, Tiny-ImageNet datasets, and uses older networks, such as ResNet, MobileNetV2, for experiments. This cannot fully demonstrate whether the proposed DFMPQ framework can be adapted to newer and more complex networks.

(3) Although the advantage of no need for re-training is obvious, it shifts the huge computational and communication costs of re-training to the training of the super network stage. I believe that training a weight-sharing mixed-precision super network is extremely difficult, but this paper focuses on the effect of the super network without elaborating on more specific details on how to train such a super network.

(4) For the baseline model, this paper only provides a unified global general training setting, which is not sufficient. I think it should provide unique hyperparameter settings and training details for each benchmark model.

---

> ### Author Rebuttal · Authors · 2026-03-31
>
> We appreciate the reviewer's careful reading and constructive feedback. We are glad that the reviewer recognizes **the key challenges DFMPQ addresses, including feature drift from coupled quantization noise and non-IID data, optimization conflicts across heterogeneous bit-widths, and the high retraining costs caused by resource changes**. We hope that our detailed response below will address your concerns and **improve your score**, which is very important to us.
>
> ## **Ans. For Q1 and W4**
>
> All baseline results in Table 1 are the best numbers reported in their original publications, where each method was trained with its complete set of stability techniques under individually optimized hyperparameters. For full transparency, the revised manuscript will add a table detailing the hyperparameter configuration of every baseline (citing public configs or standard FL defaults). Baselines with differing original protocols will be re-run under our exact unified protocol for strictly fair comparison.
>
> ---
>
> ## **Ans. For Q2**
>
> The criteria remain unchanged. In modern architectures, increasing depth adds more layers within each stage, but the number of stages stays fixed. For example, ResNet-18/50/101 all have 4 stages (differing only in blocks per stage), so crucial layers remain fixed at S=4 regardless of depth.
>
> Moreover, CKA analyses by [1] show that the "block structure" phenomenon — where intra-stage representations are highly similar — becomes more pronounced in deeper networks. This means stage output prototypes become even more reliable semantic indicators as depth increases, making SAHA's weight computation more robust rather than less. SAHA's overhead does not scale with depth, depending only on S, not total layer count.
>
> [1] Nguyen T, et al. Do wide and deep networks learn the same things? uncovering how neural network representations vary with width and depth[J]. arXiv:2010.15327.
>
> ---
>
> ## **Ans. For Q3 and W2**
>
> 1. **Applicability to Newer Architectures**
>
>    Backbones like ResNet-18, MobileNetV2, and EfficientNet-Lite0 are not outdated; they remain de facto standards for resource-constrained edge computing, encompassing representative paradigms (residual, inverted residual, compound scaling).
>
>    Furthermore, DFMPQ’s core mechanisms (CCSA, SAHA, SAGS) are model-agnostic, operating in the feature space applicable to any differentiable architecture, including hierarchical vision Transformers. Models like Swin Transformer [1] progressively reduce spatial resolution across stages, making them naturally compatible. Due to limited rebuttal time, we evaluated Swin-Transformer (Tiny-version) on Tiny-ImageNet (3 local epochs due to limited rebuttal time; other settings unchanged). The results demonstrate DFMPQ's continued effectiveness:
>
>    | Method | α=0.1 | α=0.5 | α=1   |
>    |--------|-------|-------|-------|
>    | FedAvg | 45.02 | 50.17 | 50.63 |
>    | DFMPQ  | 45.84 | 49.83 | 50.98 |
>
> 2. **Scalability to Larger Datasets**
>
>    Regarding the evaluation on larger-scale datasets, please refer to our response to **Reviewer FZrr's W2**, where we provide and discuss our new experimental results on the ImageNet dataset.
>
> [1] Liu Z, et al. Swin transformer: Hierarchical vision transformer using shifted windows. ICCV 2021.
>
> ---
>
> ## **Ans. For W1**
>
> We appreciate your concern regarding practical edge hardware deployment. Although we did not evaluate on specific physical devices (e.g., Raspberry Pi), we used hardware-agnostic metrics: BitOPs (standard proxy for latency and energy) and Model Size (proxy for memory and communication). BitOPs is highly reliable in mixed-precision quantization as it quantifies arithmetic intensity independent of hardware.
>
> To address adaptability under fluctuating hardware conditions, please refer to our response to **Reviewer FZrr (Q2 & W3)**, which demonstrates that DFMPQ can switch to new BitOPs/memory constraints within 2–9 minutes with zero retraining cost. These analyses will be included in the revised manuscript.
>
> ---
>
> ## **Ans. For W3**
>
> We agree that training a weight-sharing mixed-precision supernet has higher one-time cost than a single non-sharing model.
>
> On an RTX 5090 GPU using ResNet-18 on CIFAR-100, training the non-sharing FedAvg baseline to convergence takes approximately 2 hours and 5 minutes. Our supernet requires about 7 hours and 17 minutes under the same settings until all SAGS-extracted subnets achieve comparable performance.
>
> The major advantage is that this cost is paid only once. When the resource budget changes D times, traditional methods incur roughly 2h × D total cost, whereas DFMPQ requires only the one-time ~7-8h cost plus D × SAGS search time (3-6 minutes each). The amortized benefit becomes significant as D increases.
>
> The supernet trains within the standard FL pipeline: clients sample mixed-precision configurations each round, sharing latent weights with STE. Detailed pseudocode will be provided in the revision.

---

> > ### Author Rebuttal · Reviewer_AfS2 · 2026-04-02
> >
> > Dear authors, thank you for your responses.
> > Some of my questions have been answered, but I'm still concerned about the experimental setup on real edge devices. The experiments are on GPU servers, not the edge devices to deploy the client models (it is more like a simulation). I think this will degrade the real effectiveness of the proposed method.

---

> > > ### Author Response · Authors · 2026-04-05
> > >
> > > Dear Reviewer AfS2
> > >
> > > We sincerely appreciate the reviewer's continued attention and constructive feedback. To address the concern regarding real edge device experiments, we conducted additional experiments on a real Raspberry Pi 5 cluster. We hope the following results can resolve your concerns and dispel any doubts you may have. We would be grateful if you could consider raising the score or confidence based on these supplementary results.
> > >
> > > ## Experimental Setup
> > > We construct a real federated edge system where the clients consist of 29 Raspberry Pi 5 (8GB) nodes and the central server is a workstation equipped with a single NVIDIA RTX 4090 GPU. Each Raspberry Pi 5 features a quad-core ARM Cortex-A76 CPU (2.4 GHz), delivering approximately 30 GFLOPS (FP32) via NEON SIMD instructions, with 8 GB LPDDR4X memory. Communication is conducted over a campus WiFi LAN. We evaluate on ResNet-18 with CIFAR-10. Except that the number of participating clients is set to 29 (matching the number of available Raspberry Pi nodes), all other hyperparameters remain consistent with the main experiments.
> > >
> > > ## Training Phase Results
> > > We record the key per-round metrics during federated supernet training, as shown in the table below. Specifically, the average upload time per client includes the transmission of the SAHA module's critical-layer class-conditional prototypes and the locally updated model weights. The average download time per client covers the time required to receive the global model weights and the CCSA module's global class-conditional prototypes before local training begins. Client training time per round denotes the average time each client consumes to complete its local training in a single global round, and Aggregation time per round denotes the time the server spends calculating the aggregated weights and completing the aggregation after each global training round.
> > >
> > > | Metric                        | Value      |
> > > |-------------------------------|------------|
> > > | Avg. upload time per client   | 3.66 s    |
> > > | Avg. download time per client | 2.14 s    |
> > > | Client training time per round| ~80.2 min |
> > > | Aggregation time per round    | ~2.8 min  |
> > >
> > > ## Deployment Phase Results
> > > In real-world edge deployments, the available energy budget of each device fluctuates with battery level, thermal throttling, and power-sharing with co-located applications. Since computational energy consumption is approximately proportional to the total bit-operations[1] we adopt BitOPs as a proxy for the energy budget. We simulate five resource states corresponding to different energy constraints, ranging from 3.0 GBitOPs (power-saving mode) to 16.0 GBitOPs (normal operation), and evaluate the accuracy-efficiency trade-off achieved by SAGS under each constraint. The results are presented in the table below.
> > >
> > > | BitOPs Budget | Actual BitOPs | Model Size | Acc.   | Search Time |
> > > |---------------|---------------|------------|--------|-------------|
> > > | 3.0 G        | 2.96 G       | 2.72 MB   | 75.19% | 237 s      |
> > > | 5.0 G        | 5.00 G       | 5.00 MB   | 78.26% | 198 s      |
> > > | 8.0 G        | 7.99 G       | 6.01 MB   | 79.42% | 156 s      |
> > > | 12.0 G       | 11.95 G      | 6.40 MB   | 81.44% | 101 s      |
> > > | 16.0 G       | 15.92 G      | 6.77 MB   | 82.91% | 52 s       |
> > >
> > > Furthermore, our method offers significant advantages over retraining-based approaches in the deployment phase. Conventional methods require a complete retraining cycle (hours to days) and re-downloading the entire model (\~44 MB) whenever device resource constraints change. In contrast, DFMPQ only requires the latent weights to be downloaded once (\~2.14 s). For each subsequent resource change, the server executes SAGS to search for an optimal bit-width configuration and transmits only a 168-byte configuration list to the device (\~0.6 ms). The device then performs a deterministic local quantization conversion (\~40 ms, element-wise rounding without any gradient computation) and immediately resumes inference. This scheme reduces the transmission cost of the new configuration by **more than 260,000× (168 B vs. 44 MB)** whenever the deployment resource conditions change, and completely eliminates the retraining overhead, fully demonstrating the practical application value of DFMPQ in dynamic edge deployment scenarios.
> > >
> > > [1] van Baalen M, Kahne B, Mahurin E, et al. Simulated quantization, real power savings[C]//Proceedings of the IEEE/CVF Conference on Computer Vision and Pattern Recognition. 2022: 2757-2761.
> > >
> > > Regards
> > >
> > > The authors of the manuscript

---

### Official Review · Reviewer_4g7H · 2026-03-10

**Soundness:** 2
**Presentation:** 3
**Significance:** 3
**Originality:** 3
**Overall Recommendation:** 4
**Confidence:** 2

**Summary:**

This paper proposes a framework named DFMPQ, designed to address the issues of unstable model training and inefficient deployment arising from data heterogeneity and device heterogeneity in federated learning. The framework combines Class-Conditional Semantic Alignment (CCSA), Semantic-Aware Hybrid Aggregation (SAHA), and Sensitivity-Aware Greedy Search (SAGS) to achieve efficient edge-side model deployment without retraining.

**Compliance With Llm Reviewing Policy:**

Affirmed.

**Final Justification:**

The author has addressed my concerns.

**Key Questions For Authors:**

1. Limited number of clients in experimental setup: Whilst Table 2 explores up to 40 clients, real-world federated learning scenarios often involve hundreds or thousands of clients. Supplementary experiments with larger client scales are recommended.
2. Insufficient discussion on CCSA's privacy protection: Though the authors note CCSA ‘requires no exchange of raw data’, does broadcasting category prototypes potentially leak class distribution information? Further discussion on associated privacy risks is advised.

**Limitations:**

yes

**Strengths And Weaknesses:**

Strengths:
1. The DFMPQ framework features a clear structure, with three modules (CCSA, SAHA, SAGS) working in tandem to address challenges at different levels, demonstrating rigorous design logic.
2. The robust theoretical analysis enhances the method's credibility.
3. Extensive evaluation across multiple datasets (CIFAR-10/100, Tiny-ImageNet), varying degrees of heterogeneity, differing numbers of clients, and diverse local training rounds yields compelling experimental results.
Weaknesses:
1. Limited number of clients in experimental setup: Whilst Table 2 explores up to 40 clients, real-world federated learning scenarios often involve hundreds or thousands of clients. Supplementary experiments with larger client scales are recommended.
2. Insufficient discussion on CCSA's privacy protection: Though the authors note CCSA ‘requires no exchange of raw data’, does broadcasting category prototypes potentially leak class distribution information? Further discussion on associated privacy risks is advised.

---

> ### Author Rebuttal · Authors · 2026-03-31
>
> We sincerely thank the reviewer for their thorough evaluation and encouraging feedback. We are glad that DFMPQ is recognized as having a **clear structure with rigorous design logic**, supported by **robust theoretical analysis** and **extensive evaluation yielding compelling experimental results**. We hope that our detailed response below will address your concerns and **improve your score or confidence**, which is very important to us.
>
> ## **Ans. For Q1 & W1**
>
> We thank the reviewer for the constructive suggestion regarding scalability to larger client populations. While our main experiments (Table 2) already cover up to $N = 40$ clients — a scale commonly used in FL literature — we fully agree that real-world deployments often involve hundreds or even thousands of clients.
>
> To directly address this concern, we conducted additional experiments on CIFAR-10 and CIFAR-100 with $N = 80$ and $N = 100$ clients (under the same $\alpha = 0.5$ Dirichlet partition and all other hyperparameters as in the original Table 2). The results are summarized below:
>
> | Method | CIFAR-10 ($N=80$) | CIFAR-10 ($N=100$) | CIFAR-100 ($N=80$) | CIFAR-100 ($N=100$) |
> | :--- | :---: | :---: | :---: | :---: |
> | **FP32** | 63.34 | 57.37 | 33.44 | 23.83 |
> | **FP8** | 44.90 | 38.92 | 23.69 | 21.81 |
> | **DFMPQ** | 71.64 | 62.16 | 36.45 | 29.63 |
>
> These results demonstrate that DFMPQ consistently and significantly outperforms both FP32 and FP8 baselines even as the number of clients scales up to 100. The performance advantage remains substantial, confirming the robustness and scalability of our method to larger client populations.
>
> ---
>
> ## **Ans. For Q2 & W2**
>
> We sincerely appreciate the reviewer's concern regarding privacy. We analyze the privacy implications from both communication directions involved in our framework.
>
> * **CCSA: Server $\rightarrow$ Client.** CCSA involves only unidirectional communication from server to client. Global prototypes are computed exclusively from the server-side buffer dataset using the full-precision supernet and contain no information derived from any client's private data. The computation of local prototypes and the CCSA loss (Eq. 5) is performed entirely on the client device, without uploading any prototype to the server. Therefore, the CCSA module introduces no additional privacy risk.
> * **SAHA: Client $\rightarrow$ Server.** The SAHA aggregation strategy (Eq. 7) does require clients to upload local prototypes to the server for aggregation weight computation. However, the prototypes undergo two levels of information compression before transmission, making the privacy risk minimal in practice. First, Global Average Pooling compresses each sample's feature map from $(C, H, W)$ to a single $(C,)$ vector, completely discarding all spatial information including textures, shapes, and spatial layouts. This makes it infeasible to reconstruct any visual content of the original images from the uploaded representation. Second, class-wise averaging further aggregates all samples belonging to the same class from $(n_c, C)$ to a single $(C,)$ mean vector, where individual sample-level information is absorbed into the class-level statistic. After this dual compression, each client uploads only a highly compact class-level summary that contains no information about the number of samples in each class.

---

> > ### Author Rebuttal · Reviewer_4g7H · 2026-04-04
> >
> > Thank you for your reply. I’ll give you a higher rating.

---

> > > ### Author Response · Authors · 2026-04-04
> > >
> > > Dear Reviewer 4g7H
> > >
> > > Many thanks for your work over the past period and for higher rating.
> > >
> > > Regards
> > >
> > > The authors of the manuscript

---

### Official Review · Reviewer_FZrr · 2026-03-10

**Soundness:** 3
**Presentation:** 2
**Significance:** 2
**Originality:** 3
**Overall Recommendation:** 4
**Confidence:** 3

**Summary:**

This paper proposes DFMPQ, a federated mixed-precision quantization framework for retraining-free deployment under dynamic edge constraints. It trains a weight-sharing supernet in the federated setting and derives resource-aware subnets without further retraining. The method combines class-conditional semantic alignment, semantic-aware hybrid aggregation, and a sensitivity-aware greedy search intended to support deployment under changing resource budgets.

**Compliance With Llm Reviewing Policy:**

Affirmed.

**Final Justification:**

Thanks to the authors for the feedback and justification, and to all the reviewers for the discussions. I am glad to support a positive rating.

**Key Questions For Authors:**

- What exactly is the server-side cache dataset used by SAGS?

- Can the authors provide deployment-time evidence for dynamic adaptation?

**Limitations:**

yes

**Strengths And Weaknesses:**

### Strengths

- The paper studies a relevant problem at the intersection of federated learning and mixed-precision quantization.

- The framework is coherent, and the proposed components address plausible training and deployment issues.

- The evaluation spans multiple datasets and backbones, and the ablations provide partial support for the design.

### Weaknesses

- The main concern is a clear inconsistency between Table 1 and Table 2 under the same reported setting. For alpha = 0.5 and N = 10, DFMPQ is reported as 89.50 in Table 1 but 80.24 in Table 2 on CIFAR-10, raising doubts about the reported results or protocol alignment.

- The used datasets are limited. CIFAR-10 and CIFAR-100 are small-scale datasets; large-scale datasets, such as ImageNet, are recommended for better evaluation.

- The key claim of retraining-free deployment is not directly validated with dynamic budget switching, deployment latency, or comparison to retraining-based alternatives.

- SAGS relies on a server-side cache dataset, but the paper does not clearly explain its source, scale, or implications for the FL setting.

- The theory depends on strong assumptions that seem difficult to justify for deep nonconvex supernets, so it reads more as intuition than as a rigorous guarantee.

- The baseline comparison protocol is also not sufficiently clear, especially regarding matched resource budgets and deployment assumptions.

---

> ### Author Rebuttal · Authors · 2026-03-31
>
> We sincerely thank the reviewer for their valuable effort and constructive feedback. We are glad that DFMPQ is recognized as **addressing a relevant problem with a coherent framework and evaluation spanning multiple datasets and backbones**. We hope that our detailed response below will address your concerns and **improve your score**, which is very important to us.
>
> ## **Ans. For Q1 & W4**
>
> **Source and construction.** $D_{buffer}$ is constructed prior to client data partitioning by sampling a small balanced subset (20 images per class) from the official training set. The remaining data is Dirichlet-partitioned among clients, ensuring $D_{buffer}$ is completely disjoint from all private data.
>
> **Scale and implications for FL.** $D_{buffer}$​ is orders of magnitude smaller than aggregate client data, with negligible computational burden. Privacy implications and SAGS sensitivity to buffer size/distribution are detailed in our response to **Reviewer hzbv (Q2&W2)**.
>
> ---
>
> ## **Ans. For Q2 & W3**
>
> Mixed-precision quantization involves two stages: (1) bit-width search and (2) QAT retraining to recover accuracy. Recent methods[1] reduce stage-1 search cost but still require full per-config retraining (~1-2h for ResNet-18 on CIFAR-100 [1]), limiting dynamic adaptation.
>
> DFMPQ eliminates all per-configuration retraining. The supernet incurs a higher one-time cost (see response to Reviewer AfS2 W3), but any new config is derived via SAGS in minutes.
>
> Due to space limits, we report partial results:
>
> | Dataset       | Target | Actual | Size (MB) | Acc (%) | Search (min) |
> |---------------|--------|--------|-----------|---------|--------------|
> | CIFAR-100    | 10     | 9.89   | 4.15      | 67.37   | 2.85         |
> |              | 6      | 5.89   | 4.07      | 67.51   | 3.66         |
> | Tiny-ImageNet| 65     | 64.98  | 5.49      | 68.49   | 2.31         |
> |              | 25     | 24.75  | 4.11      | 67.59   | 9.15         |
>
> Key Conclusions: (1) Switching takes only 2–9 minutes without retraining—orders of magnitude faster than per-config retraining. (2) High performance across diverse budgets demonstrates the supernet's adaptability.
>
> [1] Li N, Su Y, Ma L. Efficient and Generalizable Mixed-Precision Quantization via Topological Entropy[C]. NeurIPS 2025.
>
> ---
>
> ## **Ans. For W1**
>
> We confirm this was a data entry error in Table 1 during manuscript preparation. The corrected results are:
>
> | Dataset        | α=0.1 | α=0.5 | α=1   |
> |----------------|-------|-------|-------|
> | CIFAR-10      | 78.86 | 80.24 | 89.27 |
> | CIFAR-100     | 62.22 | 64.15 | 66.62 |
> | Tiny-ImageNet | 64.95 | 67.29 | 68.05 |
>
> We apologize for this error and will correct it in the revised manuscript.
>
> ---
>
> ## **Ans. For W2**
>
> The majority of existing works [1][2] primarily evaluate on CIFAR-10/100 and Tiny-ImageNet, standard benchmarks for non-IID and resource-constrained FL.
>
> To address this concern, we are running ImageNet experiments with ResNet-18. Due to limited rebuttal time, only DFMPQ is completed so far: 65.07% Top-1 under avg 3-bit mixed-precision (weights and activations). Full baselines will be included in the revision.
>
> [1] Kim S W, Kim S, et al. Fedwsq: Efficient federated learning with weight standardization and distribution-aware non-uniform quantization[C]. ICCV 2025.
> [2] Chen H, Vikalo H. Mixed-precision quantization for federated learning on resource-constrained heterogeneous devices[C]. CVPR 2024.
>
> ---
>
> ## **Ans. For W5**
>
> We agree that the strong convexity assumption may not strictly hold for deep nonconvex supernets. Our analysis provides actionable insights under standard FL assumptions[1][2].
>
> Crucially, Lemma 1 establishes that the expected squared client drift admits an upper bound. This bound controls drift from quantization noise and non-IID data, keeping the analysis robust for nonconvex supernets.
>
> In the revised version, we will provide a corollary under the weaker Polyak–Łojasiewicz (PL) condition in Supp. C.3, showing that CCSA and SAHA deliver the same qualitative benefits under relaxed assumptions.
>
> [1] Reisizadeh A, et al. Fedpaq: A communication-efficient federated learning method with periodic averaging and quantization[C]. ICAIS 2020.
> [2] Karimireddy S P, et al. Scaffold: Stochastic controlled averaging for federated learning[C]. ICML 2020.
>
> ---
>
> ## **Ans. For W6**
>
> We clarify our baseline comparison protocol in two aspects:
>
> 1. **Matched Resource Budgets**
>    We strictly match on W/A bit-widths. Baselines such as AQFL and FedMPQ do not disclose exact configurations yielding their best results; we conservatively compare against those best accuracies. Even so, DFMPQ remains highly competitive at extremely low average W/A bit-widths.
>
> 2. **Deployment Assumptions**
>    All methods follow identical deployment: quantized models are deployed directly upon completion without additional fine-tuning, ensuring fair comparison of inherent performance.

---

> > ### Author Rebuttal · Reviewer_FZrr · 2026-04-04
> >
> > Thank the authors for the feedback. The feedback provided preliminary ResNet-18 ImageNet results and committed to full baseline comparisons in the revision.

---

> > > ### Author Response · Authors · 2026-04-04
> > >
> > > Dear Reviewer FZrr
> > >
> > > Many thanks for your work over the past period and for your continued support. We would like to confirm that in the revised manuscript, we will provide complete baseline comparisons on ImageNet with ResNet-18.  Furthermore, two reviewers believe that our response addresses the issues and improve the score. We still hope you can improve your score.
> > >
> > > Regards
> > >
> > > The authors of the manuscript

---

### Official Review · Reviewer_hzbv · 2026-03-12

**Soundness:** 4
**Presentation:** 4
**Significance:** 3
**Originality:** 4
**Overall Recommendation:** 4
**Confidence:** 3

**Summary:**

This paper addresses a practical challenge in federated learning: how to deploy quantized models on edge devices with changing resource limits without retraining for every setting. The proposed DFMPQ framework trains a weight-sharing mixed-precision Supernet, allowing different devices to directly extract suitable subnets for deployment. It also improves robustness during federated training by reducing feature drift, filtering noisy local updates, and enabling faster deployment search. Overall, the work is practical, well-motivated, and has clear real-world value.

**Compliance With Llm Reviewing Policy:**

Affirmed.

**Key Questions For Authors:**

1. How much extra communication overhead does CCSA introduce compared with standard FedAvg, especially when the number of classes is large?
2. How sensitive is SAGS to the quality and distribution of the server-side buffer dataset $\mathcal{D}_{buffer}$?
3. How are the “crucial layers” chosen for different model architectures, and does this require model-specific tuning?

**Limitations:**

-The potential risk of maintaining a server-side buffer and transferring class prototypes.
-The paper assumes bit-width directly translates to resource savings, but on real hardware (like DSPs or specialized NPUs), mixed-precision might not be as efficient as 8-bit fixed-point without specific kernel support. I recommend the authors add a brief discussion on how to ensure privacy when sharing prototypes and the hardware-level constraints of mixed-precision.

**Strengths And Weaknesses:**

Strengths:
- The paper provides both strong experiments and solid theoretical proofs, including convergence and gradient-based sensitivity analysis.
- Using a Supernet for retraining-free federated quantization is novel and has stronger practical value than another isolated quantization trick.
- The method fits real edge devices well, where battery and thermal conditions change and flexible bit-width switching is important.
- The paper is clearly written, easy to follow, and supported by professional figures, tables, and detailed appendix derivations.

Weaknesses:
- CCSA requires sending class prototypes from server to clients, which may reduce the overall benefit under limited networks.
- SAGS relies on buffer data at the server, raising privacy concerns and possible robustness issues under distribution shift.
- Performance is sensitive to $\lambda$, and repeated tuning for each task may reduce the method’s practicality.

---

> ### Author Rebuttal · Authors · 2026-03-31
>
> We sincerely thank the reviewer for their constructive comments and for recognizing our DFMPQ framework as a **novel, practical, and well-motivated solution** for federated quantization. We are glad that our **clear writing, strong empirical results, and theoretical guarantees** were well-received. We hope that our detailed response below will **address your concerns and improve your score or confidence**, which is very important to us.
>
> ## Answer for Q1 & W1
>
> CCSA and SAHA involve different communication directions: CCSA only broadcasts global prototypes from server to clients (no uploading required), while SAHA requires clients to upload local prototypes for aggregation weight computation (Eq. 7). Each prototype is a compact mean vector obtained via Global Average Pooling and class-wise averaging. For ResNet-18 (4 stages, channel dims 64/128/256/512, totaling 960 dims per class):
>
> | Dataset       | Classes | CCSA  | SAHA  | Total   | FedAvg  | Ratio |
> | ------------- | ------- | -------- | -------- | ------- | ------------ | ----- |
> | CIFAR-10      | 10      | ~37.5 KB | ~37.5 KB | ~75 KB  | ~93.6 MB     | 0.08% |
> | CIFAR-100     | 100     | ~375 KB  | ~375 KB  | ~750 KB | ~93.6 MB     | 0.78% |
> | Tiny-ImageNet | 200     | ~750 KB  | ~750 KB  | ~1.5 MB | ~93.6 MB     | 1.56% |
>
> Even at ImageNet scale (1,000 classes), the overhead would be \~7.5 MB (\~8% of model exchange), still modest relative to the accuracy gains. Prototype size is independent of batch size, sample count, or resolution, ensuring constant overhead.
>
>
> ## Answer for Q2 & W2
>
> **Privacy.**  $D_{buffer}$ is stored and used exclusively on the server side for prototype computation (CCSA) and sensitivity estimation (SAGS). It is never transmitted to clients and contains no client-derived information, introducing zero additional privacy risk. Holding small server-side labeled data is a standard FL assumption [1][2].
>
> **Robustness.**  To directly address the robustness concern, we conducted two sets of experiments. First, we varied buffer sizes from 5 to 50 samples per class. Second, we varied buffer distribution using Dirichlet α ∈ {0.1, 0.5, 1.0, 100}. Accuracy fluctuation is within ~1% across all settings, confirming that $\Omega_l$ (Eq. 9) captures intrinsic architectural properties rather than fitting to specific data characteristics.
>
> **Buffer size sensitivity (Acc %):**
>
> | Samples/class | CIFAR-10 | CIFAR-100 | Tiny-ImageNet |
> | --- | --- | --- | --- |
> | 5 | 89.46 | 65.53 | 66.95 |
> | 10 | 89.36 | 65.46 | 66.94 |
> | 20 (default) | 89.85 | 65.84 | 66.65 |
> | 50 | 89.08 | 65.16 | 66.61 |
>
> **Distribution shift sensitivity (Acc %):**
>
> | Buffer α | CIFAR-10 | CIFAR-100 | Tiny-ImageNet |
> | --- | --- | --- | --- |
> | 0.1 | 89.36 | 65.42 | 66.95 |
> | 0.5 | 88.76 | 65.46 | 66.54 |
> | 1 | 90.05 | 65.27 | 66.27 |
> | 100 | 89.34 | 65.16 | 66.4 |
>
> [1] Sattler F, et al. Fedaux: Leveraging unlabeled auxiliary data in federated learning[J]. IEEE TNNLS 2021.
> [2] Mai V S, et al. Federated learning with server learning: Enhancing performance for non-iid data[J]. arXiv:2210.02614.
>
> ## Answer for Q3
>
> The selection of crucial layers follows a universal, tuning-free structural rule: we designate the output layer of each network stage as a crucial layer. This allows direct identification from official architecture definitions without manual search, applied unmodified to ResNet-18, MobileNetV2, and EfficientNet-Lite0.
>
> We initially adopted existing layer-wise aggregation [1]. However, it counter-intuitively underperformed FedAvg (Table 3, Supp. A). We attribute this to intermediate layers suffering joint corruption from non-IID data bias and heterogeneous quantization noise, resulting in semantically immature features.
>
> Investigating where features become semantically reliable, CKA analysis [2][3] shows representational changes concentrate at stage boundaries, making stage outputs natural semantic consolidation points. Adjusting to stage-level aggregation achieved significant performance improvements (Table 3), with generalizability across architectures further confirmed in (Fig. 5, Supp. A).
>
> [1] Xu Y, et al. Enhancing federated learning through layer-wise aggregation over non-IID data[J]. IEEE TSC 2025.
> [2] Nguyen T, et al. Do wide and deep networks learn the same things? uncovering how neural network representations vary with width and depth[J]. arXiv:2010.15327.
> [3] Kornblith S, et al. Similarity of neural network representations revisited[C].ICML 2019.
>
> ## Answer for W3
>
> We conducted sensitivity analysis on λ of the CCSA module (Fig. 6, Supp. A), evaluating λ ∈ {0, 0.01, 0.05, 0.1, 0.2} on CIFAR-10/100 and Tiny-ImageNet (α=0.5). Results show: λ=0 yields the lowest accuracy (validating CCSA contribution); performance remains near-optimal for λ ∈ [0.05, 0.2]; and default λ=0.1 consistently achieves optimal or near-optimal accuracy on all datasets. The method is robust to λ choice **without task-specific tuning**.

---

> > ### Author Rebuttal · Reviewer_hzbv · 2026-04-02
> >
> > Thanks for your rebuttal, and I will keep my positive score.

---

> > > ### Author Response · Authors · 2026-04-02
> > >
> > > Dear Reviewer hzbv
> > >
> > > Many thanks for your work over the past period and for your continued support.
> > >
> > > Regards
> > >
> > > The authors of the manuscript

---

### Decision · Program_Chairs · 2026-04-30

**Decision:**

Accept (regular)

**Comment:**

This paper proposes deploying quantized models on edge devices with changing resource limits without retraining for every setting. The proposed DFMPQ framework trains a weight-sharing mixed-precision Supernet, allowing different devices to directly extract suitable subnets for deployment. The paper provides both strong experiments and solid theoretical proofs, including convergence and gradient-based sensitivity analysis.